# Evaluation of a Broad Panel of SARS-CoV-2 Serological Tests for Diagnostic Use

**DOI:** 10.3390/jcm10081580

**Published:** 2021-04-08

**Authors:** Maren Werner, Philip Pervan, Vivian Glück, Florian Zeman, Michael Koller, Ralph Burkhardt, Thomas Glück, Jürgen J. Wenzel, Barbara Schmidt, André Gessner, David Peterhoff

**Affiliations:** 1Institute for Medical Microbiology and Hygiene, University of Regensburg, 93053 Regensburg, Germany; maren.werner@klinik.uni-regensburg.de (M.W.); Barbara.Schmidt@klinik.uni-regensburg.de (B.S.); Andre.Gessner@klinik.uni-regensburg.de (A.G.); 2Institute for Clinical Microbiology and Hygiene, University Hospital Regensburg, 93053 Regensburg, Germany; philip.pervan@stud.uni-regensburg.de (P.P.); vivian.glueck@klinik.uni-regensburg.de (V.G.); juergen.wenzel@klinik.uni-regensburg.de (J.J.W.); 3Center for Clinical Studies, University Hospital Regensburg, 93053 Regensburg, Germany; Florian.Zeman@klinik.uni-regensburg.de (F.Z.); Michael.Koller@klinik.uni-regensburg.de (M.K.); 4Institute of Clinical Chemistry and Laboratory Medicine, University Hospital Regensburg, 93053 Regensburg, Germany; ralph.burkhardt@klinik.uni-regensburg.de; 5Kliniken Südostbayern, 83278 Traunstein/Trostberg, Germany; Thomas.Glueck@klinik.uni-regensburg.de

**Keywords:** SARS-CoV-2, COVID-19, serological immune response, serological tests, ELISA, ECLIA, LFA

## Abstract

Serological testing is crucial in detection of previous infection and in monitoring convalescent and vaccine-induced immunity. During the Severe Acute Respiratory Syndrome Corona Virus 2 (SARS-CoV-2) pandemic, numerous assay platforms have been developed and marketed for clinical use. Several studies recently compared clinical performance of a limited number of serological tests, but broad comparative evaluation is currently missing. Within this study, a panel of 161 sera from SARS-CoV-2 infected, seasonal CoV-infected and SARS-CoV-2 naïve subjects was enrolled to evaluate 16 ELISA/ECLIA-based and 16 LFA-based tests. Specificities of all ELISA/ECLIA-based assays were acceptable and generally in agreement with the providers’ specifications, but sensitivities were lower as specified. Results of the LFAs were less accurate as compared to the ELISAs, albeit with some exceptions. We found a sporadic unequal immune response for different antigens and thus recommend the use of a nucleocapsid protein (N)- and spike protein (S)-based test combination when maximal sensitivity is necessary. Finally, the quality of the immune response in terms of neutralization should be tested using S-based IgG tests.

## 1. Introduction

Serological testing is the standard diagnostic technique to detect past infection with various pathogens and is widely used to monitor vaccine-induced immunity. Immunoassays like Enzyme-linked or Electrochemiluminescence Immunosorbent Assay (ELISA/ECLIA), Lateral Flow Rapid Test (LFA), Immuno-PCR (iPCR) or Recombinant Immunofluorescence Assay (rIFA) provide methods for the detection of antigen-specific immunoglobulins (Ig) with different strengths and weaknesses [1].

To date, several unknowns remain with respect to the significance of antibody titers in the progression of Coronavirus disease 2019 (COVID-19) in the ongoing SARS-CoV-2 pandemic [2]. Whilst neutralizing antibodies can reduce viral loads, viral surface protein-directed antibodies may be of disadvantage due to mechanisms like antibody-dependent enhancement (ADE) [3,4]. Thus, careful analysis of antibody titers and their correlation with other disease parameters will help to elucidate the importance of antibodies in COVID-19 (Glueck et al., in press [5]).

Along these lines, serological testing is of utmost importance in monitoring B-cell responses to SARS-CoV-2 vaccines. Depending on the delivered antigen of a vaccine, several tests using different antigens are necessary to quantify the magnitude and longevity of vaccine-induced antibody titers. Moreover, serological assays are crucial in identifying plasma donors for therapeutic interventions and potentially in monitoring levels of therapeutic antibodies in plasma during passive immunization approaches [6].

Here, we compare 13 ELISAs/ECLIA from six different commercial providers and three in-house tests for detection and quantification of SARS-CoV-2-directed antibodies of different isotypes. Furthermore, we tested 16 LFA-based assays, which can be used in point-of-care (PoC) rapid testing or in a two-step confirmatory test scenario. As a measure of diagnostic accuracy, we determined sensitivity and specificity of the assays with respect to defined negative and positive sera. We include platform-dependent assays and easy-to-perform benchtop tests and point out advantages and disadvantages of both approaches.

## 2. Materials and Methods

Serum samples of pseudonymized donors were selected from a serum panel that we recently used to validate an in-house SARS-CoV-2 ELISA [7]. The panel consisted of five sample groups. For calculation of the test specificity, we analyzed 60 serum samples, which were collected before the SARS-CoV-2 outbreak (SARS-CoV-2 naïve group). This negative panel included 31 sera from patients tested PCR-positive for seasonal CoV with confirmed reactivity against their N-antigen (seasonal CoV group). A total of 101 sera taken during March and April 2020 from hospitalized COVID-19 patients with PCR-proven SARS-CoV-2 infection was used to calculate sensitivity for three different time points after positivity of SARS-CoV-2 reverse transcription polymerase chain reaction—(RT-PCR). Group 1 (0–5 days post PCR-positivity [dppp]; *n* = 30), group 2 (6–10 dppp; *n* = 30) and group 3 (>10 dppp; *n* = 41). This procedure was approved by the Ethical Committee of the University of Regensburg (ref. no. 20-1854-101).

Using this serum panel, we evaluated one ECLIA, 12 commercial ELISAs, three in-house ELISAs and two LFAs (Table 1). The selection of tests was based on availability in Germany as of 1st May 2020. These assays were based on recombinant virus-proteins, used either the N-antigen or S-antigen, and detected different isotypes (total Ig: *n* = 2, IgG: *n* = 6, IgA: *n* = 5, IgM: *n* = 3). All LFAs detected IgG and IgM simultaneously. For fourteen additional LFAs we used a reduced panel of 19 sera due to limited availability of tests at that time. The samples were chosen randomly within the five different sample groups. All tests were carried out according to the manufacturers’ instructions. A modification was made to our recently published ELISA protocol [7] by using a commercial substrate solution (3,3′,5,5′-Tetramethylbenzidine substrate, Mikrogen). The ELISAs were analyzed using a Microplate Reader (Model 680, Bio-Rad, Hercules, CA, USA) and the ECLIA was performed on a cobas e 801 analytical unit (Roche, Mannheim, Germany). The cutoff values and limits were calculated according to the manufacturers’ specifications. For subsequent comparative analyses the data were normalized according to the following Equation (1).
(1)$$x^{\prime }=a+\frac{\left(x-\min\left(x\right)\right)\left(b-a\right)}{\max\left(x\right)-\min\left(x\right)}$$

Here, *a* (minimal value) and *b* (maximal value) were set 0 and 0.89 for negative values, 0.9 and 1.1 for borderline and 1.01 and 10 for positive results, respectively. Max(*x*) and min(*x*) were set separately for negative, borderline and positive values according to the manufacturers’ specifications. For better interpretation and comparisons of the LFA results, we introduced a semiquantitative score depending on the color intensity of the bands (0 for negative, 1 to 3 for positive) which was assigned by the experimenter based on visual inspection. According to the manufacturers’ recommendations, any weak shade of color was counted as positive.

For calculation of the specificity and sensitivity, borderline results (based on the manufacturers’ recommendations) were regarded as negative. For correlation of assay results with virus neutralization titers (IC_50_), we used the recently published data of the 22 samples analyzed in a SARS-CoV-2 virus neutralization assay [7].

Experimental data were evaluated and plotted using GraphPad Prism (GraphPad Prism version 9.0.2 for Windows, GraphPad Software, San Diego, CA, USA). Confidence intervals were calculated using the Wilson score method [8].

## 3. Results

In this cross-sectional study, a panel of 161 sera from 161 donors was analyzed with 16 ELISAs and two LFAs (Figure 1a,b). Due to the different dynamic ranges of the S/CO values (Roche over 3 logs; ELISAs over 1–2 logs) data were normalized.

### 3.1. Determination of Specificity of the Analysed Assays

Based on the results of the SARS-CoV-2 negative panel (*n* = 60), we calculated specificities for all assays (Figure 1a, Table 2). The specificities ranged between 93.3% (Euroimmun: IgA) and 100% (Roche; Epitope: IgG, IgM; Virotech: IgA; Serion: IgG, IgA; UKR: IgG, IgM; NvM01: IgG, IgM; NvM03: IgG) and generally deviated only slightly from the manufacturers’ information. In general, IgG tests showed less false positive results (*n* = 5) compared to IgM (*n* = 4) and IgA (*n* = 7) tests given the total number of assays for these isotypes. There was no evidence of generally cross-reactive sera since out of a total of 13 sera with false positive tests only three samples (N7, N17, N23) were repeatedly tested positive with assays from different providers. Within the seasonal CoV group (*n* = 31), four assays (Bio-Rad, Virotech: IgG, Mikrogen: IgA, NvM03: IgM) produced a single false positive result (S1, S15, S3, S22).

### 3.2. SARS-CoV-2 Positive Panel

We further tested 101 serum samples from patients with PCR-confirmed SARS-CoV-2 infection. This positive panel was divided into three groups depending on the time interval of serum sample collection after positive PCR result (time point of onset of symptoms was only partially available). All SARS-CoV-2 positive groups (0–5 dppp, 6–10 dppp and >10 dppp, Figure 1b) contained several serum samples with overall low reactivity. The associated respiratory specimens shows both weak positive (2-1, 2-3, 2-6, 3-1, 3-7 with a viral load of <300 copies per mL) and clear positive RT-qPCR results (1-3, 1-4, 2-2, 2-4 and 2-5 qualitative testing; 1-8, 3-3 and 3-6 with viral loads of 6.3 × 10^7^, 1.0 × 10^4^ and 5.7 × 10^3^ copies per ml, respectively). Whether these samples were from subjects, who had cleared the infection without any seroconversion, or had seroconverted later, is unknown.

For some of the sera, a discordant reactivity towards the N- or S-antigen was apparent. This was probably a result of an individually biased immune response towards the different antigens (S: 3-4, 3-5, 3-6, 3-8; N: 1-10, 3-2, 3-10). Some sera displayed reactivity only within the IgA and/or IgM tests (1-13, 1-19, 2-7) and/or in the pan-Ig tests (Bio-Rad: 1-13, 1-19, 2-7; Roche: 1-19).

### 3.3. Determination of the Sensitivity of the Assays

The sensitivities (Table 2) ranged between 36.7% (Virotech IgM) and 87.8% (Euroimmun IgG, IgA; UKR IgG; NvM03 IgG), depending on the isotype and the different time points after positive SARS-CoV-2 RT-qPCR. In general, IgG based assays showed higher sensitivities (50%–87.8%). As expected, and in accordance with previous findings, the ratio of sera which were tested positive increased at the later time points [9]. The tests for IgM and IgA did not show such a clear finding, and the sensitivities increased or decreased within the time intervals. The pan-Ig assays showed highest sensitivities at 6–10 dppp (Bio-Rad: 86.7%) or at >10 dppp (Roche: 80.5%). However, when excluding 12 sera that showed positive results in less than two assays (*n* = 89 vs. *n* = 101), sensitivities increased and ranged between 45.8% (Virotech IgM) and 96.3% (Bio-Rad, NvM01 IgM).

Following the recommendations of the Center for Disease Control and Prevention to adapt tests to the prevalence of the target population [10], Virotech proposes two cutoff for the IgG test indices: a high prevalence cutoff for better sensitivity, and a low prevalence cutoff for better specificity in population with a predicted lower infection rate. With the latter value, the specificity within our panel improved from 96.7% to 98.3% (one false positive less), while sensitivity rose by up to 13 percent (4 additional positives in group 1, and two additional positives each in group 2 and 3) for the high prevalence cutoff.

### 3.4. Agreement between the Different Serological Assays

We then calculated the overall percent agreement for all test combinations by comparing the results of the positive panel (Figure 2). The pan-Ig and IgG tests showed the best agreement (79–97%), with overall better agreement between tests using the same antigen (N: 84–94%, S: 83–97%). The agreement for IgA and IgM tests ranged from 71–93% and 61–93%, respectively.

### 3.5. Correlation with Neutralisation Titer

Next, we correlated the assays’ results of a smaller panel of sera (*n* = 22) with previously determined SARS-CoV-2 virus neutralization titers (IC50, Figure 1c). Here, LFAs were excluded due to their semiquantitative scoring. R^2^ values ranged from 0.28 (Roche) to 0.87 (UKR: IgG, Serion: IgG). In general, the IgG tests showed the best correlation with higher values for the tests based on the S-antigen. The lower correlation coefficients of the pan-Ig tests (Roche and Bio-Rad) may be attributed to the simultaneous measurement of IgM and IgA, which showed a generally lower correlation with virus neutralization.

### 3.6. Evaluation of an Additional Panel of LFAs

Finally, we used a panel of 19 sera to evaluate 14 additional LFAs from 14 providers (Figure 1d). The tests were selected based on their availability by 1st May 2020. Three naïve sera, 4 seasonal CoV-sera (*n* = 4) and four samples for each time point were selected randomly, and for every sample we set the antibody status according to our findings with the ELISAs for each antigen. A sample was considered positive if more than 50% of the assays showed a positive result, as unclear if 50% were reactive, and as negative if less than 50% of the assays for the corresponding antigen were positive. Due to the low number of samples, sensitivity and specificity was not determined. Within the negative samples (*n* = 7), three LFAs showed a positive result, with ZECEN for two samples (N12, S15) and one sample each with Boson (N6) and Dialab (S15). Interestingly, these sera were detected false positive in several ELISAs, too. This may be attributed to crossreactive antibodies against the antigens used in these tests, or due to matrix effects. Within the positive serum panel (*n* = 12) six LFAs showed an insufficient performance with varying proportions of false negative results (ZECEN, Chemtron, RapiGEN, Türklab, AmonMed, Affimedix).

## 4. Discussion

Recently, a number of studies have compared and validated smaller panels (*n* = 1–7) of SARS-CoV-2 serological tests for diagnostic use [9,11,12,13,14,15,16,17,18,19,20,21,22,23,24,25]. In the present study, we intended to systematically cross-compare a large number of SARS-CoV-2 serological tests, and thus analyzed a broad panel of SARS-CoV-2 ECLIA-, ELISA-, and LFA-based tests which were available in May 2020, using a relatively large number of sera from patients with COVID-19, and from controls. In general, we found comparable performance of the tests in terms of specificity, which was in accordance to the manufacturers’ specifications. Sensitivities were generally lower than specified by the manufacturer. Presumably, this was due to a number of sera with reactivity below cutoff in all assays. When removing sera with reactivity in less than two tests (*n* = 12), sensitivities were comparable to the manufacturers’ specifications. By trend, a more heterogeneous picture appeared when comparing 14 LFAs, albeit a smaller panel of test-sera was used. Thus, we conclude that LFA tests should be carefully validated before being used for screening of larger cohorts. Further, verification and quantitative analysis should be performed using other assay formats. A suitable application of LFAs may be decentralized monitoring of vaccine-induced immunity, where simple PoC tests are advantageous. In such cases, the performance of assays using whole blood may be superior to serum tests due to feasibility reasons, but equivalent performance should be ensured. However, a clear limitation of LFAs is the merely semiquantitative test result.

When correlating the results of the different tests with real-virus neutralizing capacity of the sera, we could show superiority of S-based tests. This is to be expected, as the spike protein and its receptor-binding domain (RBD) in particular is the main target of neutralizing antibodies, which compete with the receptor on the cell surface. Thus, for determination of the potentially protecting quality of antibodies, an S-based test should be preferred. Furthermore IgG reactivities correlate best with neutralization in accordance with recent findings from others [26].

Serological tests can help to discriminate (the rate of) breakthrough infections after SARS-CoV-2 vaccination. The preferred test system here depends on the molecular composition of the applied vaccine. For example, the antigen of the currently applied RNA-vaccines (e.g., Comirnaty BNT162b2, by Biontech/Pfizer) is the S-protein and, thus, an active SARS-CoV-2 or past infection after vaccination can be detected by an N-based test. In regions with high seroconversion rates, prevaccination screening for already existing antibodies may save scarcely available vaccine-resources.

For some samples we found a biased immune response for one of the tested antigens. Such a constellation may be even more abundant in samples at later time points after infection, since the stability of the humoral immune response may differ between the antigens. Therefore, in a scenario where maximal sensitivity is necessary, e.g., in epidemiological studies in areas of low prevalence, a combination of S- and N-based assays with high sensitivity is recommended for screening. Subsequently, deviating, low and borderline signals may be confirmed by additional testing with highly specific serological tests or using alternative test systems like real virus or pseudotype-based neutralization assays or immune fluorescence.

Furthermore, in epidemiological follow-up studies aiming to quantify the longevity of the humoral immune response, pan-Ig assays should be avoided, since the kinetics of the different subtypes are not discriminable. In such studies, a pan-Ig test may be used for screening of positive cases and conformational, quantifying and qualifying (subtype specific response) measurements can be performed subsequently.

As a limitation of this study, mostly sera from hospitalized subjects were used due to limited availability of outpatient material. This may have biased the determination of the accuracy of the tests. Currently, the published datasets (including ours) lack independent evaluation with standard serum panels. Very recently, such reference material has been made available from the National Institute for Biological Standards and Control (NIBSC) [27]. A helpful extension of such standard sample material would be additional panels of reference material, including a number of weakly reactive sera to allow for evaluation of the precision of different tests. Along these lines, frequent independent round robin test studies are indispensable to ensure diagnostic accuracy and precision of laboratory testing.

## 5. Conclusions

In addition to nucleic acid-and antigen-testing for detection of acute infection, serological testing is able to detect past infections, and thus is essential for epidemiological surveillance and analysis of transmission patterns, and patient contacts and can help to identify asymptomatic cases. To this end, accuracy and precision of serological tests should be determined and the specifications and characteristics of the used test should be appropriate for the scenario addressed. In our study, we found superior performance of ELISAs compared with LFAs, a better correlation of S-tests with virus neutralization, and less false positive results as well as better sensitivities of IgG tests as compared to IgM and IgA tests. In our collection of sera, we identified few samples with a biased immune reaction towards the N or S-antigens. To avoid false negative testing of such cases, a combination of tests against both antigens is recommended.

Finally, information about protective antibody levels in convalescent and postvaccination sera against different emerging mutant strains is currently growing [28,29]. Serological tests with the potential to discriminate the status of protection against these novel strains will be indispensable to safeguard high-risk groups through the upcoming period of genetic drift of SARS-CoV-2, and to provide guidance for adapted vaccine strategies.

## Figures and Tables

**Figure 1 jcm-10-01580-f001:**
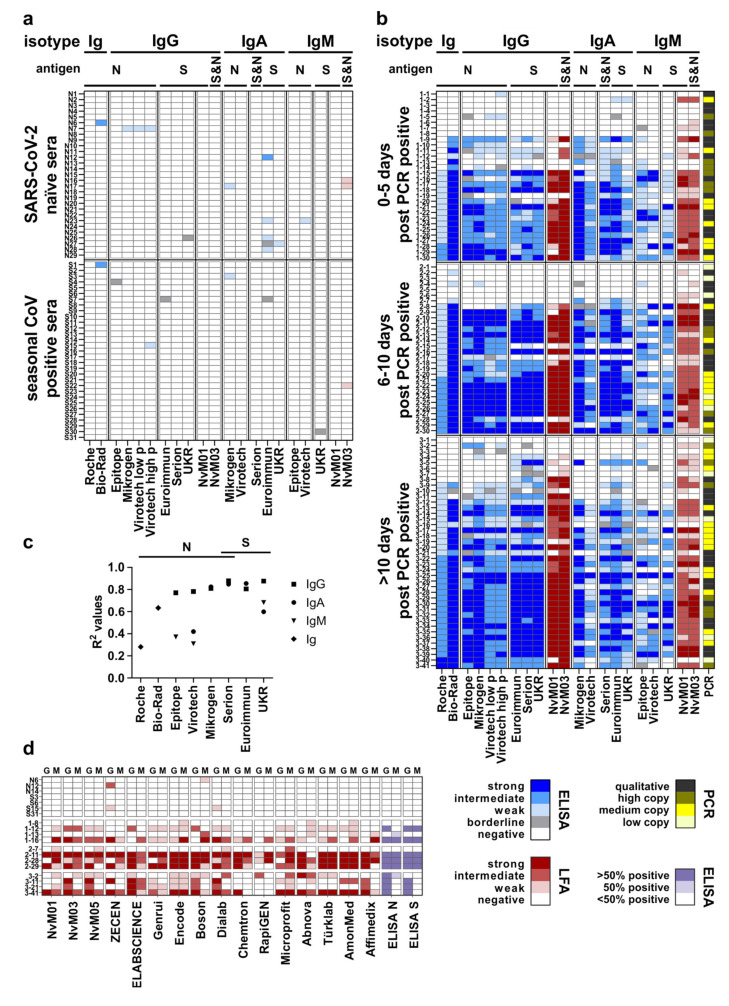
(**a**) Results from validation of the tested ELISAs and two LFAs with a panel of SARS-CoV-2 naïve and seasonal CoV-positive sera. (**b**) Assay results applying a SARS-CoV-2 PCR-positive serum panel. (**c**) Correlation of the different ELISA/ECLIA results with virus neutralization. Coefficients of determination (R^2^) were calculated and plotted for every test and detected isotype (symbols see legend). (**d**) Results from evaluation of the tested LFAs (G = IgG; M = IgM).

**Figure 2 jcm-10-01580-f002:**
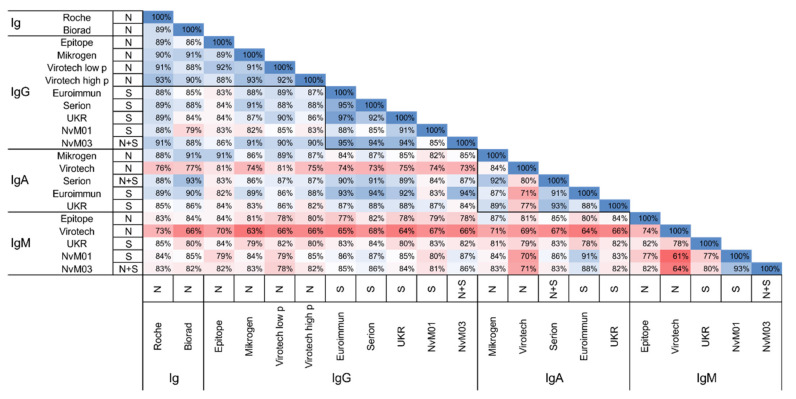
Percent agreement for all test combinations. Calculations based on the panel of positive sera. Detected antibody isotype, and used antigen of each test is given. A color gradient illustrates the percent agreement (red = low; white = in-termediate; blue = high).

**Table 1 jcm-10-01580-t001:** Tests used within this study.

Provider	Format	Product Name	Antigen	Isotype
Roche Diagnostics, Mannheim, Germany (ROCHE)	ECLIA	Cobas Ig Elecsys Anti-SARS-CoV-2	N	pan-Ig
Bio-Rad, Marnes-la-Coquette, France (Bio-Rad)	ELISA	Platelia SARS-CoV-2 Total Ab Assay	N	pan-Ig
Epitope Diagnostics, San Diego, CA, USA (Epitope)	ELISA	EDI Novel Coronavirus COVID-19 IgG, IgM ELISA Kit	N	IgG, IgM
Mikrogen GmbH, Neuried, Germany (Mikrogen)	ELISA	recomWell SARS-CoV-2 IgG, IgA	N	IgG, IgA
Virotech, Rüsselsheim, Germany (Viorotech)	ELISA	VIROTECH SARS-CoV-2 IgG, IgM, IgA ELISA	N	IgG, IgM, IgA
EUROIMMUN AG, Lübeck, Germany (Euroimmun)	ELISA	Anti-SARS-CoV-2-ELISA IgG, IgA	S	IgG, IgA
Institute Virion/Serion GmbH, Würzburg, Germany (Serion)	ELISA	Serion ELISA agile SARS-CoV-2 IgA, IgG (prototype)	IgG: S IgA: S, N	IgG, IgA
University Hospital Regensburg, Regensburg, Germany (UKR)	ELISA	SARS-CoV-2-ELISA IgG, IgM, IgA	S	IgG, IgM, IgA
Nal van Minden GmbH, Moers, Germany (NvM01, NvM03, NvM05)	LFA	NADAL COVID-19 IgG/IgM 243001	S	IgG & IgM
		NADAL COVID-19 IgG/IgM 243003	S, N	IgG & IgM
		NADAL COVID-19 IgG/IgM 243005	S, N	IgG & IgM
Zecen, Taizhou, China (ZECEN)	LFA	SARS-CoV-2 IgG/IgM Rapid Test	n.s.	IgG & IgM
Elabscience Biotechnology, Wuhan, China (ELABSCIENCE)	LFA	UNCOV-40	n.s.	IgG & IgM
Genrui Biotech, Shenzhen, China (Genrui)	LFA	2019-nCoV IgG/IgM	n.s.	IgG & IgM
Encode, Zhuhai, China (Encode)	LFA	SARS-CoV-2 IgG/IgM Rapid Test	n.s.	IgG & IgM
Boson Biotech, Xiamen, China (Boson)	LFA	Rapid 2019-nCoV IgG/IgM Test Card	n.s.	IgG & IgM
Dialab, Wr Neudorf, Austria (Dialab)	LFA	DIAQUICK Covid-19 IgG/IgM Cassette	n.s.	IgG & IgM
Chemtron Biotech Co, Shanghai, China (Chemtron)	LFA	Chemtrue 2019-nCoV IgM/IgG	n.s.	IgG & IgM
RapiGEN, Anyang, Korea (RapiGEN)	LFA	Covid-19 IgG+IgM Duo	n.s.	IgG & IgM
Microprofit Biotech Co, Shenzhen, China (Microprofit)	LFA	fluorecare SARS-CoV-2 IgG IgM	n.s.	IgG & IgM
Abnova, Taipeh, Taiwan (Abnova)	LFA	COVID-19 Human IgM/IgG Rapid Test	n.s.	IgG & IgM
Türklab, Izmir, Turkey (Türklab)	LFA	SARS-CoV-2 IgM/IgG Ab Test	n.s.	IgG & IgM
AmonMed Biotechnology; Xiamen, China (AmonMed)	LFA	COVID-19 IgM/IgG	n.s.	IgG & IgM
Affimedix, Hayward, USA (Affimedix)	LFA	TestNOW-Covid-19 IgG/IgM	n.s.	IgG & IgM

n.s. = not specified.

**Table 2 jcm-10-01580-t002:** Specificity and sensitivity of the evaluated tests.

Provider, Isotype (Antigen)	Sensitivity Manufacturer	Specificity Manufacturer	Sensitivity (CI95%); Ratio as Determined (*n* = 101)	Sensitivity (CI95%); Ratio as Determined (*n* = 89)	Specificity (CI95%); Ratio as Determined (*n* = 60)
Roche, Ig (N)	0–6 d: 65.5% 7–13 d: 88.1% ≥14 d: 100%	99.8%	0–5 d: 53.3% (36.1–69.8); 16/30 6–10 d: 76.7% (59.1–88.2); 23/30 >10 d: 80.5% (66–89.8); 33/41	66.7% (46.7–82); 16/24 85.2% (67.5–94.1); 23/27 86.8% (72.7–94.3); 33/38	100% (94–100); 60/60
Bio–Rad, Ig (N)	≤8 d: 73% 9–10 d: 100% 11–20 d: 97% 21–42 d: 100%	99.6%	0–5 d: 73.3% (55.6–85.8); 22/30 6–10 d: 86.7% (70.3–94.7); 26/30 >10 d: 75.6% (60.7–86.2); 31/41	91.7% (74.2–97.7); 22/24 96.3% (81.7–99.3); 26/27 81.6% (66.6–90.8); 31/38	96.7% (88.6–99.1); 58/60
Epitope, IgG (N)	100%	100%	0–5 d: 53.3% (36.1–69.8); 16/30 6–10 d: 73.3% (55.6–85.8); 22/30 >10 d: 70.7% (55.5–82.4); 29/41	66.7% (46.7–82); 16/24 81.5% (63.3–91.8); 22/27 76.3% (60.8–87); 29/38	100% (94–100); 60/60
Epitope, IgM (N)	45%	100%	0–5 d: 63.3% (45.5–78.1); 19/30 6–10 d: 80% (62.7–90.5); 24/30 >10 d: 58.5% (43.4–72.2); 24/41	79.2% (59.5–90.8); 19/24 88.9% (71.9–96.2); 24/27 63.2% (47.3–76.6); 24/38	100% (94–100); 60/60
Mikrogen, IgG (N)	<12 d: 86% 12–23 d: 100% >23 d: 100%	98.7%	0–5 d: 66.7% (48.8–80.8); 20/30 6–10 d: 76.7% (59.1–88.2); 23/30 >10 d: 85.4% (71.6–93.1); 35/41	83.3% (64.2–93.3); 20/24 85.2% (67.5–94.1); 23/27 92.1% (79.2–97.3); 35/38	98.3% (91.1–99.7); 59/60
Mikrogen, IgA (N)	<12 d: 50% 12–23 d: 95% >23 d: 67%	99.3%	0–5 d: 66.7% (48.8–80.8); 20/30 6–10 d: 76.7% (59.1–88.2); 23/30 >10 d: 70.7% (55.5–82.4); 29/41	83.3% (64.2–93.3); 20/24 85.2% (67.5–94.1); 23/27 76.3% (60.8–87); 29/38	96.7% (88.6–99.1); 58/60
Virotech, IgG (N)	0–5 d: 7.7% (low *p*) 6–8 d: 28.6% (low *p*) 9–11 d: 47.1% (low *p*) ≥12 d: 100% (low *p*)	100% (low *p*)	0–5 d: 60% (42.3–75.4); 18/30 6–10 d: 70% (52.1–83.3); 21/30 >10 d: 78% (63.3–88); 32/41	70.8% (50.8–85.1); 17/24 77.8% (59.2–89.4); 21/27 84.2% (69.6–92.6); 32/38	98.3% (91.1–99.7); 59/60
	0–5 d: 7.7% (high *p*) 6–8 d: 35.7% (high *p*) 9–11 d: 58.8% (high *p*) ≥12 d: 100% (high *p*)	99.8% (high *p*)	0–5 d: 73.3% (55.6–85.8); 22/30 6–10 d: 76.7% (59.1–88.2); 23/30 >10 d: 82.9% (68.7–91.5); 34/41	83.3% (64.2–93.3); 20/24 85.2% (67.5–94.1); 23/27 89.5% (75.9–95.8); 34/38	96.7% (88.6–99.1); 58/60
Virotech, IgA (N)	0–5 d: 7.7% 6–8 d: 50% 9–11 d: 64.7% ≥12 d 76.5%	100%	0–5 d: 63.3% (45.5–78.1); 19/30 6–10 d: 56.7% (39.2–72.6); 17/30 >10 d: 48.8% (34.3–63.5); 20/41	79.2% (59.5–90.8); 19/24 63% (44.2–78.5); 17/27 52.6% (37.3–67.5); 20/38	100% (94–100); 60/60
Virotech, IgM (N)	0–5 d: 0% 6–8 d: 42.9% 9–11 d: 41.2% ≥12 d 70.7%	100%	0–5 d: 36.7% (21.9–54.5); 11/30 6–10 d: 50% (33.2–66.9); 15/30 >10 d: 46.3% (32.1–61.3); 19/41	45.8% (27.9–64.9); 11/24 55.6% (37.3–72.4); 15/27 50% (34.9–65.2); 19/38	98.3% (91.1–99.7); 59/60
Euroimmun, IgG (S)	≤10 d 22.4% <10–20 d 87.5% ≥21 d 100%	99.3%	0–5 d: 56.7% (39.2–72.6); 17/30 6–10 d: 70% (52.1–83.3); 21/30 >10 d: 87.8% (74.5–94.7); 36/41	70.8% (50.8–85.1); 17/24 77.8% (59.2–89.4); 21/27 94.7% (82.7–98.5); 36/38	100% (94–100); 60/60
Euroimmun, IgA (S)	>10 d: 60.2% <10 d: 98.6%	92%	0–5 d: 70% (52.1–83.3); 21/30 6–10 d: 80% (62.7–90.5); 24/30 >10 d: 87.8% (74.5–94.7); 36/41	87.5% (69–95.7); 21/24 88.9% (71.9–96.2); 24/27 94.7% (82.7–98.5); 36/38	93.3% (84.1–97.4); 56/60
Serion, IgG (S)	96.2%	99.2%	0–5 d: 60% (42.3–75.4); 18/30 6–10 d: 76.7% (59.1–88.2); 23/30 >10 d: 82.9% (68.7–91.5); 34/41	75% (55.1–88); 18/24 85.2% (67.5–94.1); 23/27 89.5% (75.9–95.8); 34/38	100% (94–100); 60/60
Serion, IgA (S, N)	96.3%	99%	0–5 d: 70% (52.1–83.3); 21/30 6–10 d: 73.3% (55.6–85.8); 22/30 >10 d: 75.6% (60.7–86.2); 31/41	87.5% (69–95.7); 21/24 81.5% (63.3–91.8); 22/27 81.6% (66.6–90.8); 31/38	100% (94–100); 60/60
UKR, IgG (S)	>10 d: 96%	99.3%	0–5 d: 56.7% (39.2–72.6); 17/30 6–10 d: 66.7% (48.8–80.8); 20/30 >10 d: 87.8% (74.5–94.7); 36/41	70.8% (50.8–85.1); 17/24 74.1% (55.3–86.8); 20/27 94.7% (82.7–98.5); 36/38	100% (94–100); 60/60
UKR, IgA (S)	>10 d: 92%	99.3%	0–5 d: 60% (42.3–75.4); 18/30 6–10 d: 73.3% (55.6–85.8); 22/30 >10 d: 70.7% (55.5–82.4); 29/41	75% (55.1–88); 18/24 81.5% (63.3–91.8); 22/27 76.3% (60.8–87); 29/38	98.3% (91.1–99.7); 59/60
UKR, IgM (S)	>10 d: 98%	99.3%	0–5 d: 50% (33.2–66.9); 15/30 6–10 d: 66.7% (48.8–80.8); 20/30 >10 d: 58.5% (43.4–72.2); 24/41	62.5% (42.7–78.8); 15/24 74.1% (55.3–86.8); 20/27 63.2% (47.3–76.6); 24/38	100% (94–100); 60/60
NvM01 IgG (S)	97.4%	99.3%	0–5 d: 50% (33.2–66.9); 15/30 6–10 d: 63.3% (45.5–78.1); 19/30 >10 d: 78% (63.3–88); 32/41	62.5% (42.7–78.8); 15/24 70.4% (51.5–84.2); 19/27 84.2% (69.6–92.6); 32/38	100% (94–100); 60/60
NvM01 IgM (S)	86.8%	98.6%	0–5 d: 70% (52.1–83.3); 21/30 6–10 d: 86.7% (70.3–94.7); 26/30 >10 d: 85.4% (71.6–93.1); 35/41	87.5% (69–95.7); 21/24 96.3% (81.7–99.3); 26/27 92.1% (79.2–97.3); 35/38	100% (94–100); 60/60
NvM03 IgG (S, N)	98.8%	98.7%	0–5 d: 60% (42.3–75.4); 18/30 6–10 d: 76.7% (59.1–88.2); 23/30 >10 d: 87.8% (74.5–94.7); 36/41	75% (55.1–88); 18/24 85.2% (67.5–94.1); 23/27 92.1% (79.2–97.3); 35/38	100% (94–100); 60/60
NvM03 IgM (S, N)	93.7%	99.1%	0–5 d: 70% (52.1–83.3); 21/30 6–10 d: 83.3% (66.4–92.7); 25/30 >10 d: 80.5% (66–89.8); 33/41	83.3% (64.2–93.3); 20/24 92.6% (76.6–97.9); 25/27 86.8% (72.7–94.3); 33/38	95% (86.3–98.3); 57/60

Sensitivity manufacturer: d = days post symptoms onset; sensitivity as determined: d = days post positive RT-PCR; *n* = 89: 12 RT-qPCR positive sera that showed positive results in less than two assays were excluded; low/high *p*: values result from adjustment of the assay cutoff to low/high prevalence.

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
