# Peer review of "Evaluation of a Broad Panel of SARS-CoV-2 Serological Tests for Diagnostic Use"

_jcm, 2021, doi:10.3390/jcm10081580_

Round 1
Reviewer 1 Report
The manuscript that I reviewed “Evaluation of a Broad Panel of SARS-CoV-2 Serological Tests for Diagnostic Use” is a study aimed to compare sensitivity and specificity of 13 ELISAs/ECLIA from six different commercial providers, three in-house tests and 16 LFA-based assays for detection and quantification of SARS-CoV-2 directed antibodies of different isotypes, using defined negative and positive sera.
Comments
The article is very interesting and well written. I found very useful this analysis considering the necessity in this moment to obtain information about antibody titers in SARS-CoV-2 infected patients and to monitor vaccine-induced immunity. The statements described are supported by detailed presented data.
I have only two minor observations.
- I suggest to the Authors to use “n” for number and “N” for antigen N since the use of N in some cases can confuse the reader.
- I suggest the Author in the Materials and Methods section to better explain the analysis of the LFAs and the reason for which 14 LFAs were tested with a reduced panel of sera because it is not clear.
Author Response
The article is very interesting and well written. I found very useful this analysis considering the necessity in this moment to obtain information about antibody titers in SARS-CoV-2 infected patients and to monitor vaccine-induced immunity. The statements described are supported by detailed presented data.
We appreciate this positive appraisal of our manuscript.
I have only two minor observations.
I suggest to the Authors to use “n” for number and “N” for antigen N since the use of N in some cases can confuse the reader.
We have exchanged all “N” to “n” where quantities were specified.
I suggest the Author in the Materials and Methods section to better explain the analysis of the LFAs and the reason for which 14 LFAs were tested with a reduced panel of sera because it is not clear.
We have specified procedure of the analysis of the LFAs in line 89, where we now state “For better interpretation and comparisons of the LFA results we introduced a semi-quantitative score depending on the color intensity of the bands (0 for negative, 1 to 3 for positive) which was assigned by the experimenter based on visual inspection.”
At the time when we started our study, the availability of test kits was very limited in Germany. We thus went with the reduced panel of test samples. We now refer to this in the manuscript in line 75: “For fourteen additional LFAs we used a reduced panel of 19 sera due to limited availability of tests at that time.”
Reviewer 2 Report
The manuscript by Werner et al., evaluates several serological tests for diagnosing SARS-CoV2. The study is well planned with reasonable controls included. The manuscript will be stronger if authors could provide more logical reasoning of the results obtained e.g. on Page- authors noted that “in general, IgG test showed less positive results compared to IgM or IgA…..” is that expected? Brief explanation may be useful to non-immunology readers. I was surprised by non-inclusion of Abbott LFA in their panel of test since it is one of the most commonly used tests. Based on their results, authors should probably recommend what kind of tests are more reliable than others as well discuss the performance of serological test compared to genome-based tests (RT-PCR).
Author Response
The manuscript by Werner et al., evaluates several serological tests for diagnosing SARS-CoV2. The study is well planned with reasonable controls included.
We appreciate the positive feedback of Reviewer 2.
The manuscript will be stronger if authors could provide more logical reasoning of the results obtained e.g. on Page- authors noted that “in general, IgG test showed less positive results compared to IgM or IgA…..” is that expected? Brief explanation may be useful to non-immunology readers.
We can only speculate on the reasons for this finding. A generally lower serum level of IgA as compared to IgG may have caused the necessity of enhancing the sensitivity of the assays at the expense of specificity. Since we lack information about such details from the providers of the less sensitive IgA/M tests, we decided to stay at a descriptive level for this finding.
I was surprised by non-inclusion of Abbott LFA in their panel of test since it is one of the most commonly used tests.
To the best of our knowledge, Abbott has only marketed an antigen-PoC-test so far and the assays operated on Abbotts ARCHITECT or Alinity system were not available in our lab at the time, when we started our analyses.
Based on their results, authors should probably recommend what kind of tests are more reliable than others as well discuss the performance of serological test compared to genome-based tests (RT-PCR).
To point out the complimentary advantages of molecular and serological testing, we have adjusted the passage in the beginning of the Conclusion section (line 254). It now states: “In addition to nucleic acid- and antigen-testing for detection of acute infection, serological testing is able to detect past infections and thus is essential for epidemiological surveillance, analysis of transmission patterns and patient contacts and can help to identify asymptomatic cases.”
Reviewer 3 Report
This is very well done evaluation of a broad panel of SARS-CoV-2 serological tests. I have only minor suggestions for revisions.
- The Figures are too difficult read because of their small size. Consider making multiple figures out of Fig. 1.
- What were the criteria used to select the tests that were evaluated? Only a few of the assays were familiar to this US reviewer.
- Line 204. What is meant by "low reactive sera" here?
- Considering the problems with LFA do the author's really think they will find a place for patients self-monitoring of vaccine response. I think this approach would create more problems than it would solve, particularly since CDC and others are recommended against monitoring of vaccine response.
- The authors mention several times in the manuscript that labs should optimally combine sensitive N and S tests, but don't recommend how and when this should done?
Author Response
This is very well done evaluation of a broad panel of SARS-CoV-2 serological tests. I have only minor suggestions for revisions.
We also appreciate the positive feedback of reviewer 3.
The Figures are too difficult read because of their small size. Consider making multiple figures out of Fig. 1.
We have generated a new version of Fig. 1, which hopefully now supports better readability of the small parts of the figure (mainly the individual sample designations), when expanded to the full size of the page (which is done in the resubmitted version). We prefer to stay with a combined presentation of the data in figure 1 is helpful for the reader as it facilitates cross-comparison between the positive and negative control data as well as between ELISA and LFA results. Furthermore, we have expanded Fig. 2 to the full width of the page.
What were the criteria used to select the tests that were evaluated? Only a few of the assays were familiar to this US reviewer.
Main criterion of the selection of the assays (ELISA and LFA) was the commercial availability in southern Germany at the time of conceptualization of the study. We have stated this now also for the ELISAs in line 71: “The selection of tests was based on availability in Germany as of May 1st 2020.”
Line 204. What is meant by "low reactive sera" here?
We apologize for not being clear at this point. We have adapted to: “Presumably, this was due to a number of sera with reactivity below cutoff in all assays.”
Considering the problems with LFA do the author's really think they will find a place for patients self-monitoring of vaccine response. I think this approach would create more problems than it would solve, particularly since CDC and others are recommended against monitoring of vaccine response.
We fully agree with the reviewer: self-testing of serological parameters is indeed difficult to implement. Albeit we think in a later stage after vaccination in the current pandemic and depending on the stability of the induced humoral immune response decentralized determination of the serological status e.g. in specialized test stations may be necessary. Hence, we changed the sentence in line 213 to: “A suitable application of LFAs may be decentralized monitoring of vaccine-induced immunity, where simple PoC tests are advantageous.”
The authors mention several times in the manuscript that labs should optimally combine sensitive N and S tests, but don't recommend how and when this should done?
We have included the following sentences in the discussion (line 235), which clarify our opinion on a combined testing strategy: “Therefore, in a scenario, where maximal sensitivity is necessary, e.g. in epidemiological studies in areas of low prevalence, a combination of S- and N-based assays with high sensitivity is recommended for screening. Subsequently, deviating, low and borderline signals may be confirmed by additional testing with highly specific serological tests or using alternative test systems like real virus- or pseudotype-based neutralization assays or immune fluorescence.”